# Overcoming Resistance to HER2-Directed Therapies in Breast Cancer

**DOI:** 10.3390/cancers14163996

**Published:** 2022-08-18

**Authors:** Ilana Schlam, Paolo Tarantino, Sara M. Tolaney

**Affiliations:** 1Department of Hematology and Oncology, Tufts Medical Center, Boston, MA 02111, USA; 2Department of Medical Oncology, Dana-Farber Cancer Institute, Boston, MA 02215, USA; 3Division of New Drugs and Early Drug Development, European Institute of Oncology IRCCS, University of Milan, 20122 Milan, Italy

**Keywords:** HER2, resistance, breast cancer, monoclonal antibodies, tyrosine kinase inhibitors, antibody-drug conjugates

## Abstract

**Simple Summary:**

Breast cancer is the most common cancer in women in the United States. Around 15% of all breast cancers overexpress the HER2 protein. These HER2-positive tumors have been associated with aggressive behavior if left untreated. Drugs targeting HER2 have greatly improved the outcomes of patients with HER2-positive tumors in the last decades. Despite these improvements, many patients with early breast cancer have recurrences, and many with advanced disease experience progression of disease on HER2-targeted drugs, suggesting that patients can develop resistance to these medications. In this review, we summarize several mechanisms of resistance to HER2-targeted treatments. Understanding how the tumors grow despite these therapies could allow us to develop better treatment strategies to continue to improve patient outcomes.

**Abstract:**

Human epidermal growth factor receptor 2 (HER2)-positive breast cancer accounts for around 15% of all breast cancers and was historically associated with a worse prognosis compared with other breast cancer subtypes. With the development of HER2-directed therapies, the outcomes of patients with HER2-positive disease have improved dramatically; however, many patients present with de novo or acquired resistance to these therapies, which leads to early recurrences or progression of advanced disease. In this narrative review, we discuss the mechanisms of resistance to different HER2-targeted therapies, including monoclonal antibodies, small tyrosine kinase inhibitors, and antibody-drug conjugates. We review mechanisms such as impaired binding to HER2, incomplete receptor inhibition, increased signaling from other receptors, cross-talk with estrogen receptors, and PIK3CA pathway activation. We also discuss the role of the tumor immune microenvironment and HER2-heterogeneity, and the unique mechanisms of resistance to novel antibody-drug conjugates. A better understanding of these mechanisms and the potential strategies to overcome them will allow us to continue improving outcomes for patients with breast cancer.

## 1. Introduction

Breast cancer is the leading cause of cancer and the second most common cause of cancer-related mortality in women in the United States (US), accounting for over 287,000 new cancer diagnoses and over 40,000 deaths annually [1]. Human epidermal growth factor receptor-2 (HER2; erb2/neu) is overexpressed and/or amplified in around 15% of breast cancers [2,3,4]. HER2 is an oncoprotein encoded by the *HER2-neu* gene, first described in the 1980s [5]. It belongs to the epidermal growth factor receptor (EGFR) family, which is composed of four cytoplasmic membrane-anchored proteins: EGFR/HER1, HER2, HER3, and HER4 [6,7]. HER2 has no known ligand and is activated by the formation of homo- or heterodimers with other EGFR proteins, which triggers the phosphatidylinositol triphosphate kinase (PI3K) signaling pathway that controls cell growth, differentiation, and migration (Figure 1A–D) [8]. HER2+ tumors were historically associated with a worse prognosis relative to other breast cancer subtypes [2,3]. The development of HER2-directed therapies, however, has dramatically improved the prognosis of patients with HER2+ disease [3,9,10].

Despite significant improvement in the outcomes of patients with HER2+ disease, many still present with de novo or acquired resistance to HER2-targeted therapies, which leads to an early recurrence or progression of advanced disease. A better understanding of the mechanisms of resistance will allow us to continue to improve treatments and outcomes for our patients. In this narrative review, we summarize the current HER2-targeted agents, mechanisms of resistance, and potential strategies to overcome them.

## 2. HER2-Directed Therapies: Mechanism of Action and Current Indications

Multiple HER2-targeted agents are approved by the US Food and Drug Administration (FDA) and/or European Medicines Agency (EMA), including monoclonal antibodies (mAB), tyrosine kinase inhibitors (TKIs), and antibody-drug conjugates (ADC).

### 2.1. Monoclonal Antibodies

Trastuzumab is an IgG1 kappa light-chain recombinant humanized mAB that targets the extracellular domain IV of HER2 [11]. Once trastuzumab binds to HER2, it interferes with the PI3K downstream pathway and enhances antibody-dependent cellular cytotoxicity (ADCC), leading to cell death (Figure 1A) [12,13,14,15]. Trastuzumab was approved by the US FDA in 1998 in combination with chemotherapy for the treatment of metastatic HER2+ breast cancer. This approval significantly impacted the prognosis of patients with advanced HER2+ disease. Since then, its use has been expanded to the neoadjuvant and adjuvant settings.

Pertuzumab is a recombinant humanized mAB that targets the extracellular domain II of HER2 (Figure 1A), which is necessary for receptor HER2/HER3 dimerization and signaling. The addition of pertuzumab to trastuzumab and chemotherapy combination regimens has been shown to improve outcomes for patients and is now the standard of care for the (neo)adjuvant treatment of HER2+ breast cancer and the first-line therapy of advanced disease, based on the APHINITY and CLEOPATRA studies. It was approved in 2012 and 2013, respectively [10,16].

Margetuximab is a human/mouse chimeric and Fc-engineered mAB directed to HER2 [17,18]. Margetuximab and trastuzumab bind the same epitope of the HER2 receptor with different affinities; margetuximab was engineered in order to enhance the activation of the immune response while maintaining the antiproliferative effects of trastuzumab (Figure 1B) [17,18]. This drug was approved by the US FDA in December of 2020 based on the SOPHIA phase 3 trial [18,19]. In this study, margetuximab was compared to trastuzumab, both in combination with chemotherapy in heavily pretreated patients with advanced HER2+ disease [18,19]. Margetuximab led to a modest improvement in progression-free survival (PFS), which was enhanced for a subset of patients (low-affinity CD16A-F158) [18,19]. However, recent data shows that the improvement in PFS does not translate into a survival benefit [18,19]. This drug is currently being studied in the neoadjuvant setting (MARGOT trial, NCT04425018) in patients with the low-affinity genotype, and its use remains limited in current clinical practice.

### 2.2. Tyrosine Kinase Inhibitors

There are three TKIs currently approved by the US FDA for the treatment of HER2+ breast cancer: lapatinib, neratinib, and tucatinib [20,21,22,23,24]. These TKIs can inhibit HER2 and other receptors from the EGFR family; when inhibiting HER2, they prevent the phosphorylation of the tyrosine kinase residues, thus blocking downstream signaling (Figure 1C) [21,22,23,24,25]. Given their increased penetrance of the blood–brain barrier [26], TKIs are a particularly important treatment option for the prevention and treatment of brain metastases [22,23,24], which represent a major unmet need in HER2+ breast cancer.

Lapatinib is a reversible inhibitor of EGFR and HER2 [20]. The role of lapatinib in early-stage breast cancer was studied in several trials and the drug was not associated with a long-term benefit; therefore, lapatinib is not approved in that setting [27,28]. Conversely, lapatinib has been shown to be effective in the metastatic setting in combination with capecitabine in patients with pretreated HER2+ breast cancer [21], and also in combination with letrozole for patients with HER2+ and hormone receptor (HR)-positive advanced disease [29]. Lapatinib was approved in 2007 for the treatment of HER2+ advanced breast cancer.

Neratinib is an irreversible small-molecule inhibitor of EGFR, HER2, and HER4. This is the only TKI currently approved for the treatment of early breast cancer, based on the findings of the ExteNET trial [24]. In this study, patients were randomized to extended therapy with neratinib for a year vs. a placebo after the completion of standard therapy; there was an improvement in invasive disease-free survival (iDFS) in patients with node-positive and HR+ disease [24]. Importantly, patients included in the ExteNET trial were not pretreated with pertuzumab or T-DM1; thus, the clinical utility of adjuvant neratinib after treatment with these compounds remains unknown. Neratinib is also approved in the metastatic setting, based on the findings of the NALA phase 3 trial, which compared the capecitabine–lapatinib doublet to neratinib and capecitabine [23]. A major limitation in the use of this medication is its associated high rate of diarrhea, for which dose escalation and aggressive antidiarrheal prophylaxis regimens have been shown to be effective in improving drug tolerance [30]. Neratinib was approved in 2017 as an extended adjuvant therapy for early HER2+ breast cancer and in 2020, in combination with capecitabine, for the treatment of advanced disease. Pyrotinib is another EGFR, HER2, and HER4 TKI, currently only approved in China.

Tucatinib is a highly selective HER2 selective TKI. It was approved in 2020, in combination with trastuzumab and capecitabine, for patients with advanced, pretreated HER2+ disease, based on the HER2CLIMB study, which showed the superiority of this triplet compared with the doublet of capecitabine and trastuzumab [22]. This study was unique in that it included patients with untreated or progressive brain metastases, a population that is often excluded from clinical trials and represents a major unmet need. Notably, the triplet led to an improvement in the central nervous system median PFS (9.9 vs. 4.2 months-HR 0.32, 95% CI 0.22–0.48, *p* < 0.001), and the median OS was also improved in this patient population (18.1 vs 12 months-HR 0.58, 95% CI 0.40, *p* = 0.005) [26]. Tucatinib is being studied in combination with the ADC trastuzumab emtansine (T-DM1) in the post-neoadjuvant setting in the CompassHER-RD (NCT04457596) trial, and the same combination is being evaluated in the metastatic setting in HER2CLIMB-02 (NCT03975647). Additionally, the ADC trastuzumab deruxtecan (T-DXd) in combination with tucatinib is being assessed in HER2CLIMB-04 (NCT045399938) for the treatment of patients with metastatic disease.

### 2.3. Antibody–Drug Conjugates

ADCs have changed the treatment paradigm of HER2+ breast cancer. These drugs are composed of an mAB, a linker, and a cytotoxic payload [31]. T-DM1 was the first HER2-directed ADC approved for the treatment of breast cancer and was also the first ADC ever approved for the treatment of a solid tumor. It is an HER2 monoclonal antibody, with a non-cleavable linker, and the cytotoxic payload is the microtubule inhibitor emtansine [32]. It was studied in the metastatic setting in the EMILIA trial [32] and in the post neoadjuvant setting on KATHERINE [33], and was approved in 2013 and 2019, respectively.

More recently, T-DXd was approved for the treatment of HER2+ metastatic breast cancer, initially beyond the second line of therapy based on the DESTINY-Breast01 study [34] and, in March of 2022, it was approved as a second-line treatment based on the findings of the DESTINY-Breast03 study [35]. Additionally, this drug was approved by the US FDA in August of 2022 for the treatment of HER2-low metastatic breast cancer, defined as HER2 1 to 2+ on immunohistochemistry (IHC) and not amplified by in-situ hybridization (ISH) [36]. T-DXd is now recommended as a treatment option for patients with pretreated HER2-low breast cancer in the NCCN guidelines [36]. Some of the key characteristics of this T-DXd include that the HER2 mAB and the payload are conjugated through a cleavable linker; it has a high drug to antibody ratio of 8:1; the cytotoxic payload (deruxtecan, a highly potent topoisomerase I inhibitor) is membrane soluble; and, once released in the HER2+ cell, this molecule can diffuse out of the cell and can have a cytotoxic effect on surrounding HER2-tumor cells and on the tumor microenvironment (TME), a feature defined as the “bystander effect” (Figure 1D) [35].

## 3. Mechanisms of Resistance

Several mechanisms of resistance to HER2-targeted agents have been identified. In this section we discuss these mechanisms and potential strategies to overcome them.

### 3.1. Impaired Binding to HER2

One of the well-described mechanisms of resistance to HER2-targeted therapies, in particular for mABs, is impaired binding to the extracellular domain of HER2 by different mechanisms, such as structural modifications of the monoclonal antibody binding site (Figure 2A). The p95HER2 receptor is a truncated form of the full-length p185HER2 receptor, which lacks the trastuzumab binding site but retains kinase activity; this is a highly oncogenic form of HER2, as it can spontaneously form homodimers, leading to cell proliferation (Figure 2A) [37,38]. The p95HER2 receptor is present in about 30% of HER2+ breast cancers and has been found to be a poor prognostic marker, with an increased risk for nodal metastases [37,39]. Preclinical models reveal that cells expressing p95HER2 are resistant to treatment with trastuzumab monotherapy but can respond to TKIs, such as lapatinib [40]. The combination of trastuzumab and chemotherapy has been studied in this setting and, when p95HER2 expressing tumors are treated with trastuzumab and paclitaxel, the cells appear sensitive to trastuzumab, whereas combination with anthracyclines seems less effective, for reasons that still remain unclear [39]. Assessing the presence of pp95HER2 receptors is challenging; several assays have been studied, including immunofluorescence and quantitative methods, such as Western blot and flow cytometry [41,42]. These assays need to be tested in clinical trials to determine if there is a role for p95HER2 testing to guide therapeutic decisions.

Mucin-4 is a surface glycoprotein that protects epithelial structures and affects adhesion, proliferation, and the metastatic potential of cancer cells [42]. Mucin-4 stabilizes HER2, potentiating downstream pathway activation; it can also physically block the binding site for mABs, leading to resistance to these therapies [42,43]. Silencing of Mucin-4 in trastuzumab-resistant cancer cells has been shown to increase trastuzumab binding and ADCC [44]. Given that the interaction of Mucin-4 and HER2 occurs in the extracellular space, similar to the mentioned p95HER2 receptor, there is a possibility that TKIs can overcome this mechanism of resistance; this has been shown with lapatinib in preclinical models of HER2+ gastric cancer [43]. Hyaluronan is another polymer that can physically block the trastuzumab binding site-rending cells resistant to this therapy. This can be overcome by blocking the production of hyaluronan, as has been shown in in vivo and in vitro models; an alternative would be again to use TKIs to overcome this mechanism of resistance [45].

### 3.2. HER2 Mutations

Activating mutations in the *HER2/ERBB2* gene are another mechanism of resistance to HER2-directed therapies; these are found in around 2–3% of all breast cancers and are associated with a worse prognosis (Figure 2B) [46]. HER2 mutations are not detected by standard HER2 testing (IHC and ISH), as gene sequencing testing is needed to identify them. Specific mutations are associated with different prognoses and response rates to HER2-targeted therapies [47,48]. L755S is the most common mutation; it has been described as an acquired mechanism of resistance for lapatinib, with apparent crossed resistance to tucatinib, whereas it appears to respond to neratinib [47,48,49,50]. With the increasing use of next-generation sequencing in patients with metastatic breast cancer, we are likely to detect more of these mutations. HER2-mutations will need to be characterized, and additional information is needed to determine the optimal strategies to treat patients with HER2 mutations. T-DM1 and T-DXd have shown activity in HER2-mutant lung cancer [51,52], and the ongoing DPT01 basket study (NCT04639219) is assessing the activity of T-DXd in patients with solid tumors with activating HER2 mutations, including patients with advanced breast cancer.

### 3.3. Cross-Talk between HER2 and Estrogen Receptors

About 50% of HER2+ breast cancers also express the estrogen receptor (ER). HER2 and ER are key drivers of cancer proliferation in breast cancer, and preclinical models have demonstrated a correlation between these receptors in breast cancer [53]. It has been demonstrated that, in HER2+ breast cancer, HER2-directed therapies result in increased ER expression, which can lead to cell survival (Figure 2C) [54]. Based on these findings, combination therapies targeting both HER2 and ER have shown to be effective [29,55,56] However, it remains unclear if patients pretreated with HER2-targeted therapies could benefit from continuing endocrine therapy with subsequent lines of treatment, either chemotherapy or ADC-based therapies.

### 3.4. Increase Signaling from HER2 and Other Receptors

Increased expression of HER2 has been described as another mechanism of resistance to HER2-targeted agents; similarly, incomplete receptor inhibition can lead to resistance to trastuzumab. A potential strategy to overcome these is to provide HER2-targeted agents that block the receptors homogeneously and/or permanently. Additionally, other receptors can be overexpressed in HER2+ tumors, leading to resistance to cancer therapies including EGFR, HER3, and VEGF.

An HER2 blockade can be bypassed by EGFR and HER3 homodimer formation (Figure 2D); the expression of these proteins increases with trastuzumab exposure, and this has been described as another resistance mechanism [57]. In vitro studies suggested that an EGFR blockade could increase sensibility to HER2-targeted agents; however, trials with the EGFR inhibitors gefitinib and cetuximab in patients with breast cancer failed to improve patients’ long-term outcomes [57]. Notably, the TKIs neratinib and lapatinib are inhibitors of EGFR and HER2 (neratinib also blocks HER4), which have been shown to improve patient outcomes. Several HER3 blockers are being investigated as monotherapies or in combination with HER2-targeted therapies.

Another described mechanism of resistance to HER2-targeted agents is transcriptional deregulation via aberrant activation of multiple tyrosine kinases [58,59]. Preclinical models have shown that inhibition of the cyclin-dependent kinases 7 or 8 can restore sensibility to HER2-targeted agents [58,59]. Clinical studies are needed to determine the efficacy and tolerance of these medications in combination with HER2-targeted agents.

### 3.5. Downstream Pathway Activation

*PIK3CA* encodes for the p110α subunit of phosphatidylinositol 3-kinase (PI3K). When the HER2 tyrosine kinase is activated, PI3K is stimulated, which leads to activation of protein kinase B (AKT) and mammalian target of rapamycin (mTOR), leading to cell proliferation, growth, and survival [60]. Activating *PIK3CA* mutations are found in 20% of HER2+ breast cancers and have been described as a mechanism of resistance to some HER2-directed therapies [61]. These mutations have been associated with lower pathologic complete response rates in patients receiving neoadjuvant HER2-targeted therapies in combination with chemotherapy and with worse PFS for those receiving these therapies for metastatic disease [62,63]. A biomarker analysis of the CLEOPATRA study revealed that patients with wild-type *PIK3CA* have longer PFS relative to those with *PIK3CA* mutations (13.8 vs. 8.6 months in the trastuzumab plus a taxane arm, and 21.8 vs. 12.5 months in the trastuzumab plus pertuzumab and a taxane arm), suggesting that *PIK3CA* mutations are a poor prognostic marker [63]. Notably, a biomarker analysis of the EMILIA study reveals similar outcomes for patients, with or without *PIK3CA* mutations, treated with T-DM1 but worse outcomes for patients with *PIK3CA* mutations treated with lapatinib and capecitabine, suggesting that more effective therapies could potentially overcome this resistance mechanism [64].

Another potential strategy to overcome HER2 resistance is by blocking the PI3K pathway at the PI3K, AKT, or mTOR levels [65]. Several HER2-targeted agents inhibit this pathway as a part of their anti-cancer activity [65]. The role of the mTOR inhibitor everolimus was assessed in the BOLERO1 [66] and BOLERO 3 trials [67]. In the latter, everolimus led to a modest improvement in PFS but with significant toxicities [67,68]. Neoadjuvant therapy with trastuzumab, paclitaxel, and the PI3K inhibitor buparlisib was then assessed in the NeoPHOEBE trial: the study was discontinued due to significant toxicities [69]. PI3K inhibitors are also being studied in HER2+ breast cancer: EPIK-B2 (NCT04208178) is a phase 3 study assessing the role of the PI3K-alpha inhibitor alpelisib in combination with trastuzumab and pertuzumab as maintenance for patients with HER2+, PIK3CA mutant advanced breast cancer. Additionally, mutation-specific PIK3 inhibitors such as LOXO-783, which targets PI3K-alpha H1047R, a mutation present in around 15% of breast cancers, are being developed and may prove to be beneficial with further study.

*PTEN* is an important tumor-suppressor gene; it inhibits the PI3K pathway and plays an important role in DNA repair [70,71]. Loss of *PTEN* due to mutations has been reported in 5–10% of all breast cancers, while loss of heterozygosity has been reported in 40–50% of cases [72,73]. *PTEN* mutation rates vary by breast cancer subtype, with around 5% in the HER2+ subtype, 11% in HR+, and up to 35% in triple-negative cancers [74]. *PTEN* and *PIK3CA* alterations are mutually exclusive in most cases; however, concomitant mutations have been reported [75]. The clinical implications of *PTEN*-loss in patients with HER2+ breast cancer remain unclear. An analysis of the BCIRG 005 and 006 trials assessed the outcomes of patients with early-stage HER2+ disease—with or without *PTEN*-loss—treated with chemotherapy and trastuzumab; it revealed that patients with *PTEN*-loss still benefit from trastuzumab but have overall worse PFS and OS outcomes [74]. Similar results were reported in an analysis of the APHINITY trial [76]. Overall, these data suggest that *PTEN* is a prognostic but not a predictive biomarker [74,76]. Conversely, tumors with increased PI3K activity have low *PTEN* levels and respond better to everolimus-containing regimes [77]. Further research is needed to determine if other inhibitor combinations are safe and can improve patient outcomes. Finally, *PTEN* mutations promote MAPK pathway dependency in breast cancer [78]; the combination of PI3K and MEK inhibitors have been shown to improve responses to HER2+ gastric cancer [79], and future research is needed to determine if this combination could improve outcomes in patients with HER2+ breast cancer.

The JAK/STAT3 pathway has also been studied in HER2+ breast cancer and has shown that patients with activation of this pathway have better outcomes relative to those without pathway activation [80]. A study showed that STAT-associated gene signatures were predictive of trastuzumab resistance in patients being treated in the adjuvant setting, suggesting that a combination of agents targeting the STAT pathway could improve the response to trastuzumab [81]. However, the combination of the JAK2 inhibitor ruxolitinib with trastuzumab for the treatment of patients with pretreated metastatic disease was studied in a phase 1/2 clinical trial and showed no improvement of outcomes, compared to historical controls. Additional research is needed to determine if there is a role for other JAK/STAT targeting agents, and the optimal setting [82].

### 3.6. Failure to Trigger ADCC

ADCC was initially described as a mechanism of action of trastuzumab in animal models [83], and then confirmed in a small clinical study that showed a strong lymphocytic infiltrate in all patients treated with neoadjuvant trastuzumab, particularly in those who achieved a pCR [84]. Several factors can affect the ability of HER2-directed therapies to trigger ADCC; some examples include Fcγ receptor polymorphisms and the quantity of tumor-infiltrating lymphocytes (TILs). Several strategies have been studied to enhance ADCC associated to HER2-targeted therapies.

Three Fcγ receptors expressed on immune cells regulate ADCC: CD16A, CD32A, and CD32B [85]. Margetuximab was engineered to retain the epitope specificity of trastuzumab; however, five amino acids were modified to increase affinity for the activating Fcγ receptor CD16A and to decrease activity to the inhibitory Fcγ receptor CD32B, with the goal to enhance ADCC [17,18]. Margetuximab was compared with trastuzumab, both in combination with chemotherapy in patients with previously treated HER2+ advanced breast cancer in the mentioned phase 3 SOPHIA study [18] (Figure 1B).

Immune checkpoint inhibitors (ICIs) have reshaped the treatment paradigm of multiple malignancies, including triple-negative breast cancer [86,87,88]. Preclinical models have demonstrated that ICIs can enhance and improve the therapeutic activity of HER2-targeted agents [89]. However, the benefit of adding immunotherapy to HER2-targeted therapies in clinical trials has been limited. PANACEA was a single-arm, phase 1b-2 study in which patients with trastuzumab-resistant advanced breast cancer were treated with the programmed cell death 1 (PD1) inhibitor pembrolizumab in combination with trastuzumab [90]. A total of 58 patients were enrolled in this study, six of the 40 patients with programmed cell death ligand-1 (PDL1)-positive disease achieved an objective response, and there was resistance in the PDL1-population; none of the 12 patients in this group had a clinical response [90]. With a 13-month median follow-up in the PDL1+ population, the PFS was 2.7 months, and the OS was not reached [90]. In the PDL1 population, with a median follow-up of 12 months, the PFS was 2.5 months and the OS was 7 months [90]. KATE2 was a randomized, phase 2 trial in which 202 patients with previously treated advanced HER2+ breast cancer were randomized to T-DM1, with or without atezolizumab [91]. While the addition of atezolizumab did not improve PFS in the intention-to-treat (ITT) population, there was a modest benefit in the PDL1+ population, although the clinical significance remains uncertain, and patients treated with ICI had increased toxicities [91]. The ongoing phase 3 KATE3/MP42319 (NCT04740918) trial is randomizing patients with HER2+ and PD-L1+ advanced breast cancer to T-DM1, with or without atezolizumab in the second-line setting, while NCT03199885 is assessing the role of atezolizumab in combination with a taxane, trastuzumab and pertuzumab in the first-line setting in a PD-L1 unselected patient population. These studies are going to determine if there is a role for ICIs in advanced HER2+ breast cancer. The addition of ICIs to standard therapies has also been studied in the neoadjuvant setting. In the IMpassion050 phase 3 trial, 454 patients with early HER2+ breast cancer were randomized to receive a standard anthracycline-containing neoadjuvant regimen, trastuzumab and pertuzumab with or without atezolizumab [92,93]. Patients who were randomized to atezolizumab continued it in combination with trastuzumab and pertuzumab in the post-neoadjuvant setting [92,93]. IMpassion050 was discontinued due to an unfavorable benefit–risk profile; with five fatal adverse events in the atezolizumab arm [93]. The addition of ICI did not improve pCR rates in the ITT or PDL1+ populations, and the one-year event-free survival was also similar in both groups [93]. The ongoing Astefania trial (NCT04873362) is randomizing patients to receive standard T-DM1 therapy, with or without atezolizumab, for patients with high-risk, HER2+ breast cancer and residual disease after neoadjuvant therapy. Based on these findings, ICIs are currently not used for the treatment of HER2+ breast cancer; additional information is needed to determine if a subgroup of patients could benefit from the addition of immunotherapy. PDL1 expression appeared to be a predictor of response in patients with metastatic disease, similar to what has been shown in triple-negative breast cancer; however, these studies were not designed to assess uniquely this population and the ICI was used beyond first-line, which has been associated with modest responses in triple-negative disease [94]. PDL1 was not predictive of response in IMpassion050, again consistent with what has been reported in early triple-negative disease [88]. High quantities of TILs have been associated with improved outcomes in early and advanced HER2+ breast cancer [95,96]; this biomarker has not been studied as a predictor of response to ICI in HER2+ breast cancer.

In summary, it has been well established that ADCC plays a key role in the mechanism of action of trastuzumab, and it is possible that immune escape is a mechanism of resistance to HER2-targeted therapies. However, strategies such as modifying the trastuzumab Fcγ receptor or adding ICI to standard therapies have shown a modest improvement in patient outcomes to date.

### 3.7. HER2 Heterogeneity and Level of HER2 Expression

Intratumoral heterogeneity of HER2 expression impacts the response to HER2-targeted therapies and has been described as another mechanism of resistance [97,98]. HER2 heterogeneity was described in a recent study as an area of HER2 amplification in more than 5% but less than 50% of tumor cells, or a HER2 negative area by ISH [97]. This study assessed 164 HER2+ tumors that were treated with neoadjuvant T-DM1 and pertuzumab [97]. Heterogeneity was identified in 10% of cases; notably the pathologic complete response rate in patients without HER2-heterogeneity was 55%, compared to 0% in those with heterogeneous tumors [97]. These findings underscore the critical role of HER2 heterogeneity in clinical outcomes; future research is needed to determine how to optimize treatment selections based on HER2 expression.

The level of HER2 expression can also impact the response to HER2-targeted agents. Based on the current guidelines, HER2+ is defined as IHC results of 3+ or 2+ with amplification by ISH [99]. Around 50% of breast cancers have some HER2 expression; however, they do not meet the criteria for HER2 positivity and have been defined as “HER2-low” [100]. Several trials have studied the use of HER2-targeted agents for HER2-low breast cancer. The phase 3, randomized NSABP B47 trial assessed the role of adjuvant chemotherapy with or without trastuzumab in patients with early-stage HER2-low breast cancer [101]. A total of 3270 patients were randomized, including 565 (17.3%) with HR− disease; the addition of trastuzumab did not improve outcomes for patients with HER2-low breast cancer [101]. Based on these findings and previous data, HER2-targeted mABs are not used for the treatment of HER2-low disease.

More recently, the ADC T-DXd, a more potent HER2-directed therapy with a bystander effect, was studied for the treatment of HER2-low advanced breast cancer. DAISY was a phase 2 trial in which 179 patients with heavily pretreated advanced breast cancer were treated with T-DXd; the study included 68 patients with HER2 3+ breast cancer, 73 with HER2-low breast cancer, and 38 with HER2-null breast cancer (IHC 0) [102]. At a median follow-up of 15 months, patients in the HER2+ cohort had a response rate of 70%; the median duration of response was 9.7 months and the mPFS 11.1 months [102]. In the HER2-low cohort, the response rate was 37.5%; the median duration of response was 7.6 months, and the mPFS was 6.7 months (6.9 in the HR+ and 3.5 in the HR−). Notably, responses were also reported in the HER2-null cohort, with a response rate of 29.7%, a duration of response of 6.8 months, and an mPFS of 4.2 months (4.5 in the HR+ and 2.1 in the HR−) (102). DESTINY-Breast04 was a phase 3 trial, which assessed the role of T-DXd vs. chemotherapy of the physician’s choice in 557 patients with pretreated metastatic HER2-low disease (494 HR+, 63 HR−) [36]. In the HR+ cohort, the mPFS was 10.1 vs. 5.4 months, favoring the ADC (hazard ratio = 0.51, *p* ≤ 0.001), and the OS was 23.9 vs. 17.5 months, again favoring T-DXd (hazard ratio 0.64, *p* = 0.003) [36]. In the ITT population, the mPFS was 9.9 vs. 5.1 months, favoring the ADC (hazard ratio = 0.50, *p* ≤ 0.001), and OS was 23.4 vs. 16.8 months, favoring T-DXd (hazard ratio 0.64, *p* = 0.001) [36].

The findings of DESTINY-Breast04 changed the treatment paradigm of advanced breast cancer, and T-DXd is now a new option for a large number of patients with advanced disease (Figure 2E); however, several questions remain [103]. DESTINY-Breast04 included a small number of patients with HR−/HER2-low breast cancer and additional data are needed to confirm the findings of this study. IHC is qualitative, and the implementation of quantitative HER2-assessment methods is being studied to determine if it yields more reliable and reproducible results in improving the prognostic and predictive yield of this biomarker [104]. Some of the studied quantitative methods include a dual-antibody, proximity-based approach (HERmark) [105] and artificial intelligence-based imaging [106]. Although these are currently part of clinical practice, it is possible that, in the near future, we will incorporate quantitative HER2 analysis to guide clinical decisions. In summary, the use of potent HER2-targeted agents may overcome the resistance to HER2-targeted agents related to heterogeneity, which has been a long-term challenge.

### 3.8. Unique Mechanisms of Resistance to ADC

ADCs have reshaped the treatment of breast cancer; however, patients eventually develop resistance to these medications, which presents as a recurrence or progression of the disease. The resistance to these therapies in HER2+ breast cancer can partially be explained by the mechanisms described above, as the ADC is composed of an mAB that needs to bind to the extracellular domain of HER2 in order to have an effect; however, unique mechanisms of resistance to ADCs have been also described (Figure 2F).

Increased activity and expression of the drug efflux pump is a well-known mechanism of resistance to chemotherapy. This has also been reported as an acquired mechanism of resistance to T-DM1 [107]; notably, the inhibition of these pumps restored T-DM1 sensitivity [108]. ADCs with non-cleavable linkers require proteolytic degradation in the lysosome to release the cytotoxic payload, which then needs to be transferred by a lysosome to its intracellular target [109]. Variation in lysosomal transport has been reported as a mechanism of acquired resistance to ADCs with non-cleavable linkers, such as T-DM1 [110]. This could potentially be overcome by treating patients with an ADC with a similar mAB and a cleavable linker.

The mechanisms of resistance to ADCs have been studied in patients with TNBC treated with sacituzumab govitecan (SG) and have been found to be associated with the target or the payload [110]. The level of target expression required for ADC function is unclear; as mentioned above, patients with HER2-low expression benefited from the ADC T-DXd [102]. Similarly, patients with TNBC and low TROP2 expression have shown to benefit from SG [111]. Frameshift mutations in *TOP1* and mutations in *TACSTD2/TROP2* (the latter leads to diminished cell surface binding by antibodies, such as SG) have been described as mechanisms of acquired resistance to SG and are other examples of target alterations [110]. The potency of the payload may play a role and payloads with more bystander effect may work even in the setting of lower target expression. Payload sensitivity of the tumor type and of the patient’s tumor based on prior therapies should also be considered, as these can play a role in drug resistance [112].

New and promising ADCs are being developed, including bispecific ADC, ADC with immune-stimulating payloads, and radionuclide ADCs [31]. Combination therapies are also being studied to increase ADCC, assess target expression, and improve drug internalization of ADCs. An example includes a combination of ADCs and ICIs to enhance activity and ADCC [113], such as the mentioned KATE2 and KATE3/MP42319 (NCT04740918) trials combining T-DM1 and an ICI [91]. Another example is a phase 1, single-arm study (NCT03523572) that combined T-DXd with nivolumab in 52 patients with pretreated HER2+ and HER2-low breast cancer [114]. This study showed antitumor activity consistent with prior studies with T-DXd; however, it is unclear if the ICI improved outcomes, and a randomized study is needed to answer this question [114]. Additionally, ICIs have shown limited efficacy in pretreated patients with breast cancer; it is also unclear if there could be a role for this combination beyond first-line therapy [115].

In terms of target expression, a preclinical study showed that, after exposure to T-DM1, the level of HER2 expression decreased in cancer cells [107]. It is possible that the use of a more potent ADC with a bystander effect could overcome this mechanism of resistance; however, there is a paucity of data on the sequencing of ADCs and cross-resistance [102].

ADC efficacy requires endocytosis into the target cell [116]. Various ADCs, including T-DM1, enter the cell via clathrin-mediated endocytosis [116]. Preclinical models have shown that, when T-DM1 enters the cell via caveolin-mediated endocytosis instead, the efficacy of the drug decreases as the ADC is internalized into caveolin-1 vesicles, decreasing lysosomal activity [117]. This has been reported in T-DM1 only, and it is possible that using an ADC with a cleavable linker could overcome this resistance mechanism.

## 4. Conclusions

HER2-targeted therapies have changed the outcomes of patients with HER2+ and, more recently, HER2-low breast cancer. Nonetheless, de novo or acquired resistance to these targeted therapies remains a challenge. Multiple potential mechanisms of resistance to HER2-directed therapies have been described; developing a deeper understanding of these mechanisms and potential strategies to overcome them is crucial to improving outcomes for patients with breast cancer.

## Figures and Tables

**Figure 1 cancers-14-03996-f001:**
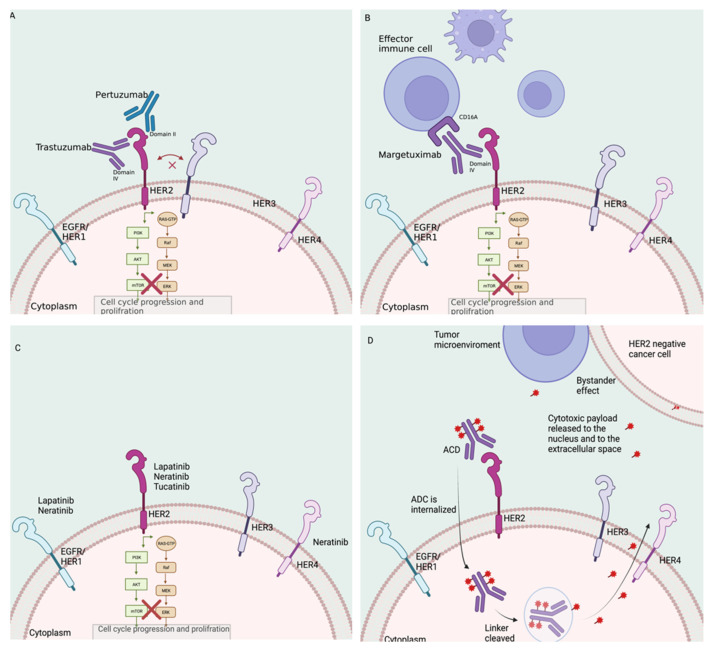
Mechanism of action of HER2-targeted therapies. Legend: (**A**) mechanism of action of trastuzumab and pertuzumab; (**B**) mechanism of action of margetuximab; (**C**) mechanism of action of tyrosine kinase inhibitors; (**D**) mechanism of actions of antibody-drug conjugates with a cleavable linker (Created with BioRender).

**Figure 2 cancers-14-03996-f002:**
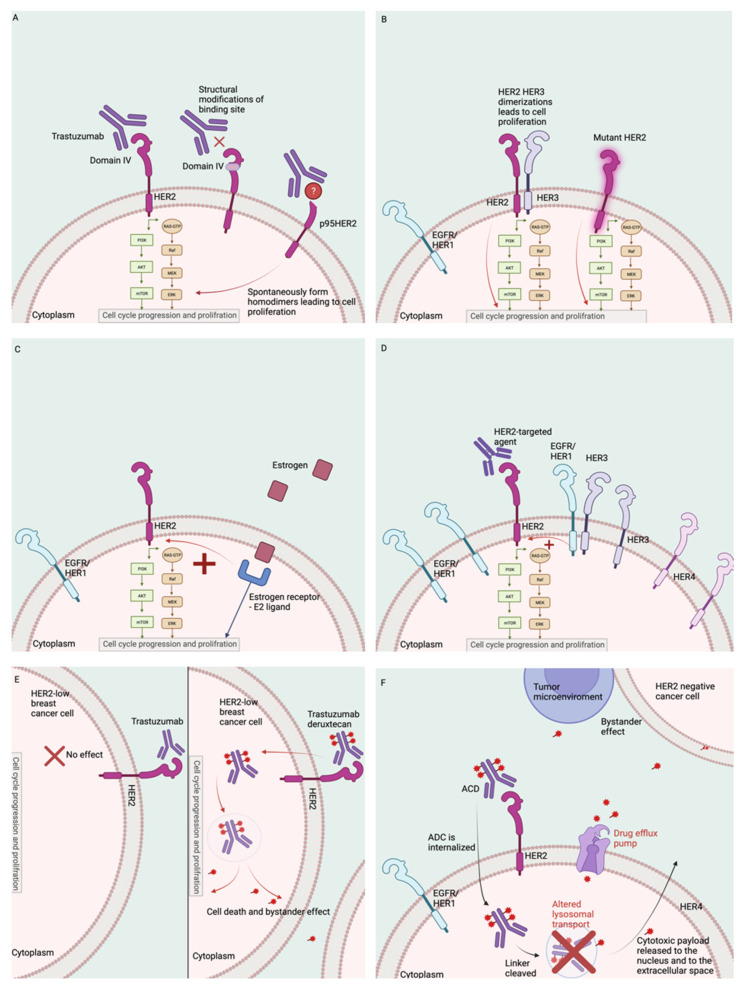
Mechanisms of resistance to HER2-targeted agents. Legend: (**A**) structural modifications of the monoclonal antibody binding site; (**B**) activating mutations in the HER2/ERBB2 gene; (**C**) cross-talk between HER2 and estrogen receptors; (**D**) increased expression of EGFR and HER3. (**E**) The level of HER2 expression can impact response to certain HER2-targeted therapies; this figure shows that HER2-low tumors do not usually respond to monoclonal antibodies but can respond to antibody-drug conjugates. (**F**) Increased activity and expression of the drug efflux pump and altered lysosomal transport are unique mechanisms of resistance to antibody-drug conjugates.

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
