# Peer review of "Overcoming Resistance to HER2-Directed Therapies in Breast Cancer"

_cancers, 2022, doi:10.3390/cancers14163996_

Round 1

Reviewer 1 Report

In this comprehensive review authors discuss the mechanisms of resistance to HER2-targeted therapies, including monoclonal antibodies, small tyrosine kinase inhibitors, and antibody-drug conjugates. Authors evaluate mechanisms of impaired binding to HER2, incomplete receptor inhibition, increased signaling from alternative receptors, crosstalk with estrogen receptors, and PIK3CA pathway activation. They also discuss the role of the tumor immune microenvironment and HER2-heterogeneity and the unique mechanisms of resistance to novel antibody-drug conjugates. Understanding of these mechanisms and the potential strategies to overcome them will help to improve outcomes for our patients. Despite the broad coverage of therapies and resistance mechanisms, there are important newer advances that are not covered in this review. Many of the varied mechanisms of resistance to HER2-targeting drugs involve transcriptional reprogramming associated with constitutive activation of signaling pathways parallel or downstream of HER2 (C. Vernieri et al., 2019 doi.org/10.1016/j.critrevonc.2019.05.001).  Resistance of breast cancer to HER2 inhibitors involves reprogramming of the kinome through HER2/HER3 signaling via the activation of multiple tyrosine kinases and transcriptional upregulation. Development of resistance to both conventional and targeted drugs frequently starts with non-genetic transcriptional changes that allow tumor cells to adapt and survive drug exposure, as summarized in a recent paper in Nature Rev. Cancer (Marine et al., 2020 DOI: 10.1038/s41568-020-00302-4). Recent articles have shown role of transcription regulating kinases CDK7 and CDK8/19 in resistance to HER2 inhibition (Ding et al., 2022 DOI: 10.1073/pnas.2201073119; Sun et al., Oncogene 2020 doi: DOI: 10.1038/s41388-019-0953-9). Hopefully, these suggestions will help, and this review will be enhanced by overviewing above-mentioned aspects of the cancer resistance mechanisms.

Author Response

Reviewer 1

  • Despite the broad coverage of therapies and resistance mechanisms, there are important newer advances that are not covered in this review. Many of the varied mechanisms of resistance to HER2-targeting drugs involve transcriptional reprogramming associated with constitutive activation of signaling pathways parallel or downstream of HER2 (C. Vernieri et al., 2019 doi.org/10.1016/j.critrevonc.2019.05.001).

Pages 12 and 13: Thank you for this comment and the reference. We summarized the downstream pathway activation pathways as a mechanism of resistance on page 12 and 13. The following text and the reference provided was added “Another potential strategy to overcome HER2 resistance is by blocking the PI3K pathway at the PI3K, AKT or mTOR levels (65). Several HER2-targeted agents inhibit this pathway as a part of their anti-cancer activity (65)”

  • Resistance of breast cancer to HER2 inhibitors involves reprogramming of the kinome through HER2/HER3 signaling via the activation of multiple tyrosine kinases and transcriptional upregulation. Recent articles have shown role of transcription regulating kinases CDK7 and CDK8/19 in resistance to HER2 inhibition (Ding et al., 2022 DOI: 10.1073/pnas.2201073119; Sun et al., Oncogene 2020 doi: DOI: 10.1038/s41388-019-0953-9)

Page 12: thank you for this comment, the following paragraph was added “Another described mechanism of resistance to HER2-targeted agents is transcriptional deregulation via aberrant activation of multiple tyrosine kinases. Preclinical models have shown that inhibition of the cyclin-dependent kinases 7 or 8 can restore sensibility to HER2-targeted agents. Clinical studies are needed to determine the efficacy and tolerance of these medications in combination with HER2-targeted agents”

  • Development of resistance to both conventional and targeted drugs frequently starts with non-genetic transcriptional changes that allow tumor cells to adapt and survive drug exposure, as summarized in a recent paper in Nature Rev. Cancer (Marine et al., 2020 DOI: 10.1038/s41568-020-00302-4).

Page 16, 17: Thank you so much for this comment and for sharing with us this interesting paper. We discuss tumor heterogeneity on pages 16 and 17, and we included this reference in the discussion of tumor heterogeneity. More data are needed to determine is non-genetic transcriptional changes lead to tumor progression and drug resistance in HER2 positive breast cancer.

Reviewer 2 Report

The authors present a narrative review focusing on HER2 resistance mechanisms. The manuscript is well written including relevant figures.

However, one must have in mind that absence of objective and systematic selection criteria in review methods might result in a number of methodological shortcomings leading to potential bias of the author's interpretation and conclusions. The authors have not addressed this important issue?

A few additional comments:

Introduction:  The authors state that “HER2 positive breast cancer accounts for app. 20% of all breast cancer”. This number is too high and is not in accordance with daily clinical practice. Today HER2 positive breast cancer accounts for 10-15% ie: J Pathol Clin Res. 2018 Oct; 4(4): 262–273.

Mechanisms of resistance: The authors describe various mechanisms of impaired binding to Her2 hereby p95HER2 but without describing the challenges associated with determination of for instance p95HER2 by IHC in clinical practice?

Her2 heterogeneity and level of Her2 expression: The authors very briefly mention the potential of other methods for HER2 determination – this important topic needs additional focus in the manuscript.

Author Response

Reviewer 2

  • One must have in mind that absence of objective and systematic selection criteria in review methods might result in a number of methodological shortcomings leading to potential bias of the author's interpretation and conclusions. The authors have not addressed this important issue?
    Thank you for this comment, we opted to write a narrative review to be able to provide a broader scope about mechanisms of resistance to HER2-targeted therapies. We were not trying to answer a specific question and considered a narrative review to be more appropriate.

  • The authors state that “HER2 positive breast cancer accounts for app. 20% of all breast cancer”. This number is too high and is not in accordance with daily clinical practice. Today HER2 positive breast cancer accounts for 10-15% ie: J Pathol Clin Res. 2018 Oct; 4(4): 262–273.

Pages 3, 4: Abstract and introduction have been updated, stating that HER2-positive cancers account for 15% of all breast cancer

  • Mechanisms of resistance: The authors describe various mechanisms of impaired binding to Her2 hereby p95HER2 but without describing the challenges associated with determination of for instance p95HER2 by IHC in clinical practice?

Page 9: The following test was added “Assessing the presence of pp95HER2 receptors is challenging; several assays have been studied, including immunofluorescence and quantitative methods -such as western blot and flow cytometry. These assays need to be tested in clinical trials to determine if there is a role for p95HER2 testing to guide therapeutic decisions”

  • Her2 heterogeneity and level of Her2 expression: The authors very briefly mention the potential of other methods for HER2 determination – this important topic needs additional focus in the manuscript.

Page 18: the following text was added “Some of the studied quantitative methods include a dual-antibody, proximity based approach (HERmark) (98) and artificial intelligence-based imaging (99). Although these are currently part of the clinical practice, it is possible that in the near future, we will incorporate quantitative HER2 analysis to guide clinical decisions.”

Reviewer 3 Report

The manuscript entitled “Overcoming resistance to HER2-directed therapies in breast cancer’’ submitted by Ilana Schlam et al, for the special issue Resistance in Breast Cancer discusses the mechanisms of resistance to different HER2-targeted therapies, including monoclonal antibodies, small tyrosine kinase inhibitors, and antibody-drug conjugates.

The study results demonstrate that mechanisms such as impaired binding to HER2, incomplete receptor inhibition, increased signaling from other receptors, cross-talk with estrogen receptors, and PIK3CA pathway activation. They also discuss the role of the tumor immune microenvironment and HER2-heterogeneity and the unique mechanisms of resistance to novel antibody-drug conjugates.

 Authors concluded by saying that understanding of the mechanisms and the potential strategies to overcome will continue to improve outcomes for the patients.

The authors have reported various studies, in the manuscript but there are certain things that need to be corrected and some needs the addition of literature.

General comments:

Ø  Authors have have discussed HER2-directed therapies, their mechanism of action and current indications. Also Multiple HER2-targeted agents have been approved by the US Food and Drug Administration (FDA) and/or European Medicines Agency (EMA), including monoclonal antibodies (mAB), tyrosine kinase inhibitors (TKIs), and antibody-drug conjugates (ADC).But I wonder whether authors have discussed nano-particle based drug therapy in this context.

Ø  How about the role of mutations in other genes, I mean genetic cause of Her2 related phenomenon?

Ø   

Ø  Authors have shown Mechanisms of resistance to HER2 targeted agents, but I wonder if authors can discuss about the role of ERK, STAT and AKT pathways as well.

Ø  Please discuss properly Cell death and bystander effect

Author Response

Reviewer 3

  • Authors have discussed HER2-directed therapies, their mechanism of action and current indications. Also Multiple HER2-targeted agents have been approved by the US Food and Drug Administration (FDA) and/or European Medicines Agency (EMA), including monoclonal antibodies (mAB), tyrosine kinase inhibitors (TKIs), and antibody-drug conjugates (ADC).But I wonder whether authors have discussed nano-particle-based drug therapy in this context.

Thank you for this comment, we decided to include only groups of drugs that are currently approved in clinical practice.

  • How about the role of mutations in other genes, I mean genetic cause of Her2 related phenomenon?

Thank you for this comment, we discussed PIK3CA mutations on pages 12 and 13

  • Authors have shown Mechanisms of resistance to HER2 targeted agents, but I wonder if authors can discuss about the role of ERK, STAT and AKT pathways as well.

Page 14: the following text was added “Finally, PTEN mutations promote MAPK pathway dependency in breast cancer (77), the combination of PI3K and MEK inhibitors have shown to improve responses to HER2+ gastric cancer (78), and future research is needed to determine if this combination could improve outcomes in patients with HER2+ breast cancer”

Page 14: the following test was added “The JAK/STAT3 pathway has also been studied in HER2+ breast cancer and has shown that patients with activation of this pathway have better outcomes relative to those without pathway activation (79). A study showed that STAT-associated gene signatures were predictive of trastuzumab resistance in patients being treated in the adjuvant setting, suggesting that a combination of agents targeting the STAT pathway could improve response to trastuzumab (80). However, the combination of the JAK2 inhibitor ruxolitinib with trastuzumab for the treatment of patients with pretreated metastatic disease was studied in a phase 1/2 clinical trial and showed no improvement of outcomes compared to historical controls. Additional research is needed to determine if there is a role for other JAK/STAT targeting agents and the optimal setting (81)”

Pages 12-13: we discussed the role of the PI3K and its effect on AKT

  • Please discuss properly Cell death and bystander effect

Page 8 “the following text discusses bystander effect: “Some of the key characteristics of this T-DXd include that the HER2 mAB and the payload are conjugated through a cleavable linker, it has a high drug to antibody ratio of 8:1,  the cytotoxic payload (deruxtecan, a highly potent topoisomerase I inhibitor) is membrane soluble, and once released in the HER2+ cell, this molecule can diffuse out of the cell and can have a cytotoxic effect on surrounding HER2- tumor cells and on the tumor microenvironment (TME), a feature defined “bystander effect”.”

We also updated the indication for the use of TDX-d, which was approved las week for the treatment of HER2-low disease (page 8: Additionally, this drug was approved by the US FDA  in August of 2022  for the treatment of HER2-low metastatic breast cancer, defined as HER2 1 to 2+ on immunohistochemistry (IHC) and not amplified by in-situ hybridization (ISH) (35))

Round 2

Reviewer 3 Report

The revised manuscript entitled “Overcoming resistance to HER2-directed therapies in breast cancer’’ submitted by Ilana Schlam et al, for the special issue Resistance in Breast Cancer discusses the mechanisms of resistance to different HER2-targeted therapies, including monoclonal antibodies, small tyrosine kinase inhibitors, and antibody-drug conjugates.

Authors have revised few sections based on the comments raised by the reviewers

I wonder if authors would discuss some Combination therapies that are being assessed to increase ADCC, target expression, and drug internalization of ADC in terms of HER2 and breast cancer.

The authors present a clear and comprehensive review of on HER2 and Breast cancer therapies. The figures and tables are appropriate and easy to interpret. The statements and conclusions are coherent.

Overall, this is an interesting study, however there are some important queries which are mentioned in the general comments to Authors, needs to be addressed before the final acceptance of the manuscript.

Also, Typo’s need to be corrected and to discuss the Combination therapies that are being assessed to increase ADCC, and target expression to catch the bigger audience.

English needs to be corrected.

Author Response

Reviewer 3

I wonder if the authors would discuss some Combination therapies that are being assessed to increase ADCC, target expression, and drug internalization of ADC in terms of HER2 and breast cancer.

Thank you for this comment. The following text was added to pages 20-21: New and promising ADCs are being developed including bispecific ADC, ADC with immune-stimulating payloads, and radionuclide ADCs. Combination therapies are also being studied to increase ADCC, assess target expression, and improve drug internalization of ADC. An example includes a combination of ADCs and ICIs to enhance activity and ADCC, such as the mentioned KATE2 and KATE3/MP42319 (NCT04740918) trials combining T-DM1 and an ICI (91). Another example is a phase 1, single-arm study (NCT03523572) that combined T-DXd with nivolumab in 52 patients with pretreated HER2+ and HER2 low breast cancer (113).  This study showed antitumor activity consistent with prior studies with T-DXd; however, it is unclear if the ICI improved outcomes and a randomized study is needed to answer this question (113). Additionally, ICIs have shown limited efficacy in pretreated patients with breast cancer, it is also unclear if there could be a role for this combination beyond first-line therapy (114).

In terms of target expression, a preclinical study showed that after exposure to T-DM1, the level of HER2 expression decreased in cancer cells (107).  It is possible that the use of a more potent ADC with bystander effect could overcome this mechanism of resistance; however, there is a paucity of data on sequencing of ADCs and cross-resistance (103). The use of TKIs could also be explored in this setting.

ADC efficacy requires endocytosis into the target cell (115). Various ADCs, including T-DM1, enter the cell via clathrin-mediated endocytosis (115). Preclinical models have shown that when T-DM1 enters the cell via caveolin-mediated endocytosis instead, the efficacy of the drug decreases as the ADC is internalized into caveolin-1 vesicles decreasing lysosomal activity (116). This has been reported in T-DM1 only and it is possible that using an ADC with a cleavable linker could overcome this resistance mechanism.

Typos need to be corrected, and English needs to be corrected

Thank you for this comment. The paper was carefully reviewed, and typos and English were corrected.
